biochemistry/evolution/molecular biology

proteins, fishes, collagen, phylogenetics, Bayesian, molecular clock

**Author for correspondence:**
Virginia L. Harvey
e-mail: virginia@palaeome.org

 

# Phylogenetic analyses of ray-finned fishes (Actinopterygii) using collagen type I protein sequences

Virginia L. Harvey[1,2], Joseph N. Keating[3] and Michael Buckley[1,2]

[1]Department of Earth and Environmental Sciences, School of Natural Sciences, University of Manchester, Manchester M13 9PL, UK
[2]Manchester Institute of Biotechnology, University of Manchester, 131 Princess Street, Manchester M1 7DN, UK
[3]School of Earth Sciences, University of Bristol, Life Sciences Building, Tyndall Avenue, Bristol BS8 1TQ, UK

VLH, 0000-0003-0796-8287; MB, 0000-0002-4166-8213

Ray-finned fishes (Actinopterygii) are the largest and most diverse group of vertebrates, comprising over half of all living vertebrate species. Phylogenetic relationships between ray-finned fishes have historically pivoted on the study of morphology, which has notoriously failed to resolve higher order relationships, such as within the percomorphs. More recently, comprehensive genomic analyses have provided further resolution of actinopterygian phylogeny, including higher order relationships. Such analyses are rightfully regarded as the 'gold standard' for phylogenetics. However, DNA retrieval requires modern or well-preserved tissue and is less likely to be preserved in archaeological or fossil specimens. By contrast, some proteins, such as collagen, are phylogenetically informative and can survive into deep time. Here, we test the utility of collagen type I amino acid sequences for phylogenetic estimation of ray-finned fishes. We estimate topology using Bayesian approaches and compare the congruence of our estimated trees with published genomic phylogenies. Furthermore, we apply a Bayesian molecular clock approach and compare estimated divergence dates with previously published genomic clock analyses. Our collagen-derived trees exhibit 77% of node positions as congruent with recent genomic-derived trees, with the majority of discrepancies occurring in higher order node positions, almost exclusively within the Percomorpha. Our molecular clock trees present divergence times that are fairly comparable with genomic-based phylogenetic analyses. We estimate the mean node age of Actinopteri at ~293 million years (Ma), the base of Teleostei at ~211 Ma and the radiation of percomorphs beginning at ~141 Ma

(~350 Ma, ~250–283 Ma and ~120–133 Ma in genomic trees, respectively). Finally, we show that the average rate of collagen (I) sequence evolution is 0.9 amino acid substitutions for every million years of divergence, with the $\alpha$3 (I) sequence evolving the fastest, followed by the $\alpha$2 (I) chain. This is the quickest rate known for any vertebrate group. We demonstrate that phylogenetic analyses using collagen type I amino acid sequences generate tangible signals for actinopterygians that are highly congruent with recent genomic-level studies. However, there is limited congruence within percomorphs, perhaps due to clade-specific functional constraints acting upon collagen sequences. Our results provide important insights for future phylogenetic analyses incorporating extinct actinopterygian species via collagen (I) sequencing.

# 1. Introduction

Ray-finned fishes (Superclass Actinopterygii) include approximately 34 000 described species [1] and exhibit substantial diversity in both morphology and ecology as a consequence of their approximately 400 million years of evolution [2]. Understanding their phylogenetic relationships allows us to consider the evolutionary processes responsible for generating such diversity. The first large-scale reorganization of fish classification came in 1966 [3], with a comprehensive morphological study of modern teleost fishes, generating a classification that still resembles teleostean systematics today [4]. Prior to this, the most widely accepted general classification was published by Berg [5], from which the order suffix of '-formes' has been retained [4]. By 1989, progress from the last two decades of research into fish phylogenetics was summarized by Nelson [6], who concluded that early relationships and branching patterns in the fish tree of life were by then relatively well understood. However, a great challenge remained in resolving the relationships between percomorphs, which he referred to as 'the bush at the top' problem [6, p. 325–336]. By the 1990s, methods to establish higher order phylogenetic relationships in fishes were progressing towards the use of molecular characters, including analyses of ribosomal RNA [7] and complete mitochondrial sequences [8], with huge contributions to phylogenetics and evolution in all the major fish lineages (except agnathans) by 2015 [9]. Over the past few decades, greater capacity for research and larger sets of nuclear gene markers became available, paving the way for multilocus (e.g. [10]) and eventually multigene phylogenetic analyses (e.g. [11]). Such expertise has developed greater consensus for a number of major lineages, such as the branching organization at the base of teleosts and the radiation of percomorphs with greater than 17 000 species [2,4,12].

As a whole, genomic-based analyses represent the 'gold standard' in phylogenetic research. However, congruence can sometimes be strengthened with a combination of different data partitions, such as the inclusion of morphological data to molecular analyses [13–15], or the combined analysis of nuclear and mitochondrial DNA sequences [16,17]. Furthermore, where genomic data are unavailable, such as in extinct taxa, the use of surviving proteins in bone deposits may be the only remaining molecular recourse to inferring phylogenetic relationships (e.g. [18]). Proteins contain transcribed genetic information and have a preservation potential that is an order of magnitude greater than that of DNA [19,20], occasionally surviving for millions of years and into geological time scales [21,22]. Protein analysis for phylogenetics can also be cheaper than the analysis of DNA and is less prone to contamination as it omits an amplification stage [23].

The protein collagen type I (hereafter 'collagen (I)'), the most dominant protein in vertebrate bone, combines qualities of stability, longevity and amino acid sequence diversity, which have previously proven useful for phylogenetic reconstructions in extinct mammal taxa [24–26]. The triple-helical molecule of the collagen (I) chain is formed of two identical alpha-1 ($\alpha$1 (I)) chains (gene COL1A1a) and one chemically dissimilar alpha-2 ($\alpha$2 (I)) amino acid chain (gene COL1A2) in most vertebrates, in an '$(\alpha1)_2\alpha2_1$' arrangement. However, it is present in an alternative heterotrimer, '$\alpha1_1\alpha2_1\alpha3_1$', in certain fish species [27], with the latter arrangement containing an alpha-3 ($\alpha$3 (I)) chain (gene COL1A1b). Structurally, each of the three collagen (I) chains comprise repeating triplicates of (Gly-Xaa-Yaa)$n$, in which glycine (Gly) occupies every third residue position, and 'Xaa' and 'Yaa' are most often occupied by proline (Pro) and its hydroxylated form, hydroxyproline (Hyp). Despite this repetitive nature, collagen (I) amino acid sequences (hereafter referred to as 'collagen (I) sequences') have demonstrated species-specific discrimination, such as between salmon (*Salmo salar*) and trout (*S. trutta*) [28]. The $\alpha$2 (I) protein chain in all observed vertebrates is more variable in its sequence than $\alpha$1 (I), which is thought to have occurred due to its reduced requirement for Pro and Hyp (Pro + Hyp)

[29]—a factor that could, in theory, promote an increase in the rate of sequence evolution for this chain. Of the three protein chains, $\alpha 3$ (I) shows higher levels of sequence variation still and is suggested to be faster evolving than either $\alpha 1$ (I) or $\alpha 2$ (I) [30], despite its origin from the duplication of the COL1A1a gene near the time of adaptive radiation of bony fish [31,32]. By all accounts, the $\alpha 3$ (I) chain, which was first reported in 1965 [27], has been heavily understudied [33] with a reasonable breadth of knowledge but little depth with regard to its prevalence. The $\alpha 3$ (I) chain is known to be widely distributed across numerous fish orders, including Acipenseriformes, Anguilliformes and some but not all Salmoniformes, but has only been assessed in a total of 34 species to date (see [31] for review). The chain is also reported to be tissue specific [34] and displays changeable expression between different environments [35].

It is currently understood that the degree of collagen (I) sequence disparity between different taxa in the vertebrate kingdom can be coarsely correlated with relative divergence time (see [30] for review). However, the vast majority of research conducted on collagen (I) sequences to date has been on mammals, with phylogenies generally based on incomplete sequence data via bottom-up proteomics (e.g. [26,36]). To date, no study has specifically analysed the dependability of collagen-based phylogenetics for fishes, nor comprehensively assessed the rate of collagen (I) sequence evolution in any taxa. As collagen (I) has successfully been extracted from archaeological specimens dating to at least the Pliocene (approximately 3.5 Ma) [21] and with a growing number of collagen-based phylogenies in the literature, an underlying understanding of collagen (I) sequence evolution and its application to phylogenetics is now essential. This is not a suggestion to replace genetic-based trees, but rather to define whether collagen-based trees are reliable approximations, particularly where genetic preservation may be lacking, such as in the fossil record. To address this, we analysed 67 published modern fish collagen (I) sequences from 60 different species throughout a range of taxonomic orders extracted from the NCBI online protein search tool, BLASTP and Ensembl. Our primary objective was to compare collagen-derived topological and time-calibrated phylogenies with published molecular-based trees to investigate the utility of collagen (I) for generating robust phylogenetic signal in fishes. Our secondary objective was to use molecular clock models to assess amino acid substitution rates for each of the collagen (I) $\alpha 1$, $\alpha 2$ and $\alpha 3$ chains in fishes. Our study also includes a technical assessment of collagen (I) sequences in fishes, a necessity for enabling sequence concatenation. The dataset for this primary foray into fish phylogenetics using collagen (I) focuses on published amino acid sequences for extant fish species and does not contain extinct species, but the results of this study will be able to inform the direction of future analyses in this regard.

# 2. Methods

All collagen (I) amino acid sequences available for actinopterygians (ray-finned fish species; $n = 56$), chondrichthyans (cartilaginous fish species; $n = 3$) and sarcopterygians (lobe-fin fish species; $n = 1$) (electronic supplementary material, table S1) were extracted from the NCBI online search tool 'Protein BLAST' [37] (Basic Local Alignment Search Tool for proteins; BLASTP) and Ensembl 2016 [38], and compiled into a custom sequence database (electronic supplementary material, table S2). Sequences were recorded under the requirement that both $\alpha 1$ (I) (1058 amino acids) and $\alpha 2$ (I) (1041 amino acids) chains were available per species; with $\alpha 3$ (I) sequences (1062 amino acids), where available, compiled only for species with $\alpha 1$ (I) and $\alpha 2$ (I) chains. Sequences that were listed as 'low-quality proteins' or had not been subject to 'final NCBI review' were ignored. The custom database was then opened in sequence alignment software BioEdit (v. 7.2.5) and manually concatenated and aligned (see Collagen (I) sequences: technical overview). Where the $\alpha 3$ (I) sequence was absent for a particular species, the alignment was completed with dashes (-), under the assumption that the chain is absent from the species rather than from the online search tools.

## 2.1. Phylogenetics

### 2.1.1. Bayesian topology analysis

Bayesian phylogenetic trees were estimated using MrBayes software (v. 3.2.7) [39]. A collagen (I) sequence dataset of 49 taxa, removing duplicate sequences so that each taxon is represented by a single set of genes, was run in PartitionFinder2 [40] using the 'MrBayes only' option and testing both linked and unlinked branch lengths (electronic supplementary material, Information). Both branch

length options supported Dayhoff [41] as the best fitting model of analysis. This model selection is further supported by its recognized suitability for modelling amino acid substitutions in other studies [42] and its use in previous collagen-based phylogenies (e.g. [24]). To assess topology, phylogenetic trees were estimated under the 'linked' criterion, whereby the model assumes that each α-chain partition shares a set of underlying branch lengths. We ran the analysis for 15 million generations, sampling every 1500 (nchains = 4, temp = 0.1, samplefreq = 1500). The first 25% of trees were discarded as burnin. We computed both the 50% Majority Rule Consensus (MRC) tree and Maximum Clade Credibility (MCC) trees. For this and all subsequent Bayesian analyses, we used Tracer v. 1.7.1 software [43] to check convergence. Analyses were considered converged if ESS scores were greater than or equal to 200 for each parameter in each independent trace, as well as in the combined trace. Topology was assessed through comparison with comprehensive genomic phylogenies from Betancur-R *et al.* [4] and Hughes *et al.* [2]. Trees were pruned in R to include the maximum number of taxa common to all sets of data (i.e. both collagen and genomic datasets) (*n* = 28). The trees were then drawn in the program Mesquite [44] and compared in R using the function comparePhylo() in the 'ape' package [45].

### 2.1.2. Tree space visualization

To understand the distribution of phylogenetic signal between the three collagen (I) α-chain data partitions, we used tree space visualization. Tree space was visualized in R using a custom script (see electronic supplementary material, Information), using the phylogenetic packages 'phangorn' [46], 'Quartet' [47], 'vegan' [48] and 'ade4' [49]. The quartets metric has a number of benefits over other distance metrics. In particular, it is less biased by rogue taxa [50]. Prior to analysis, taxa without an α3 (I) chain were removed (total taxa = 42). We then analysed the pruned dataset using both linked and unlinked topologies across partitions. The linked analysis was run under the parameters described above. The unlinked analysis was performed using the 'unlink topology(all)' and 'unlink Brlens(all)' options for 150 million generations, sampling every 15 000 generations. For computational efficiency, we sampled 200 random post-burnin trees from each α-chain partition-specific topology and 200 random post-burnin trees from the linked partitioned analysis. For our tree space visualization, we also included the consensus tree from Hughes *et al.* [2], pruned to include the same taxa as in the partition-specific and linked topologies. All trees were unrooted. We calculated a tree to tree distance matrix, obtaining the quartet distances between every tree in the combined sample and applied Cailliez's transformation to ensure the resulting distance matrix was Euclidean [51,52]. We visualized the resulting distance matrix using both classical multidimensional scaling (MDS) using 'phangorn' [46] and nonmetric multidimensional scaling (NMDS) using 'vegan'. The results were plotted using the packages 'ggplot2' [53] and 'cowplot' [54].

### 2.1.3. Bayesian clock analysis

The rate of sequence evolution per collagen (I) α-chain was investigated in MrBayes using both a uniform and birth–death clock prior (see electronic supplementary material, Information). We used the tree topology from Hughes *et al.* [2], as the most recent and comprehensive phylogeny available, and applied eight soft fossil calibrations taken from Benton *et al.* [55] (electronic supplementary material, table S3). We ran each analysis for 25 million generations, sampling every 2500 generations. The analysis was partitioned per collagen (I) α-chain (α1, α2 and α3). We applied an independent gamma rates (IGR) relaxed clock model, whereby each branch is assigned an independent rate derived from a gamma distribution. The resulting timetree was plotted using the MCMC.tree.plot() function in MCMCtreeR [56]. Estimated evolutionary rates of amino acid substitutions for collagen (I) were calculated as follows: the mean clockrate (in substitutions per site per million years) was obtained from the '.pstat' file output by MrBayes, as were the relative partition rates. We multiplied the mean clockrate by the relative partition rate to give the rate in substitutions per site per million years. We then multiplied this number by the number of sites in the partition to generate the mutation rate (in substitutions per million years). Divergence dates inferred from the collagen (I) molecular clock analysis were compared to those of published studies from Betancur-R *et al.* [4] and Hughes *et al.* [2]. Trees were rooted with a chondrichthyan outgroup comprising *Scyliorhinus* (Scyliorhinidae), *Rhincodon* (Rhincodontidae) and *Callorhinchus* (Callorhinchidae).

## 2.2. Collagen (I) sequences: technical overview

Through the process of studying the collagen (I) sequences for this research, we have uncovered a number of technical insights into fish collagen (I) that are valuable to discuss here. Collagen (I) has a highly conserved amino acid sequence, considering the ~400 million years of divergence time between the chondrichthyan and osteichthyan lineages. This level of conservatism is maintained in part by the mandatory glycine (Gly; G) residues that are positioned in almost every third location, but conservatism is also shown in two further residues, arginine (Arg; R) and lysine (Lys; K), which are highly conserved across all vertebrates, and together categorize the α-chains into 'tryptic peptides', referencing where a tryptic enzyme would cleave [23] (electronic supplementary material, table S4). Of particular interest in this study is the fact that there were two sequence variations listed as 'α1' (I) on BLASTP for most species of fish, each of which did not have a labelled α3 (I) chain (46 of 60). These two 'α1' variations differ from one another by approximately 25–29% within the same species. In-depth comparison of these chains with the five published and labelled α3 (I) sequences (from *Anguilla japonica, Carassius auratus, Danio rerio, Gadus morhua* and *Oncorhynchus mykiss*) has allowed the separation of these variants into two groups: 'true' α1 (I) chains and α3 (I) chains that we find here to have been misidentified/mislabelled as α1 (I). This mislabelling, albeit misleading in its inaccuracy, is not altogether unsurprising given that the genes that code for α1 (I) and α3 (I) have very similar terminologies—COL1A1a and COL1A1b, respectively—referencing the understanding that COL1A1b originated following a duplication of the COL1A1a gene [31,32]. Following our sequence comparison, we propose the following key α-chain discriminator that can be applied to separate α1 (I) from α3 (I) helical domains—through the sequence associated with collagen (I) tryptic peptides 32 and 33 (T32/33; or α1 334–350 following nomenclature in [57]). In α1 (I) chains, the sequence is present as two distinct tryptic peptides, separated by a lysine residue (K) in the ninth position; for example in zebrafish (*Danio rerio*), this peptide (α1 334–350) is GGPGVVGP**K**GATGEPGR. In α3 (I) chains, the sequence for T32/33 (α3 334–350) becomes one tryptic peptide, with the K residue substituted for glutamine (Q); thus in zebrafish, this peptide has the sequence GANGPMGA**Q**GASGESGR. This rule is unanimous across all the fish species in our custom database ($n = 60$). To firmly test this, we aligned full procollagen (I) sequences, including signal peptides, propeptides, telopeptides and the triple-helical domain, for the 'true' α1 (I) and suspected α3 (I) amino acid chains from a selection of species in our database (*Anguilla japonica, Esox lucius, Ictalurus punctatus, Danio rerio, Oreochromis niloticus* and *Seriola dumerii*). We observed the residue at the 1264th position to be Cys (C) in α1 (I) and Ser (S) in α3 (I), following Saito *et al.* [58] who used this trend to separate α1 (I) from α3 (I) chains in *Anguilla japonica* and *Oncorhynchus mykiss*. Only 8% of species in this study (5 of 60) had correctly labelled α3 (I) chains in BLASTP. By distinguishing between the α1 (I) and α3 (I) chains, we uncovered α3 (I) sequences in a further 77% of species ($n = 46$), which brings the total number expressing the $\alpha 1_1 \alpha 2_1 \alpha 3_1$ arrangement to 85% (51 of 60) (electronic supplementary material, table S1). As this study represents a highly diverse taxonomic range, including osteichthyans (20 orders) and chondrichthyans (3 orders), representing millions of years of evolution, it is likely that these rules of separation will be applicable to collagen (I) sequences in all fish species and not just those in our database. Distinguishing the collagen (I) chains in this way has been vital for the concatenation of full sequences for the phylogenetic analyses herein.

Of interest, the α3 (I) chain appears limited but not universal to the teleosts in our analysis, with 94% having an available sequence. This chain is, however, apparently absent from two of the five genera of cyprinids (Cypriniformes: Cyprinidae), and from herring, *Clupea harengus* (Clupeiformes: Clupeidae). Similar trends were shown by Kimura [31] analysing fish skin collagen (I), whereby the majority of teleosts species (19 of 25) had the more complex $\alpha 1_1 \alpha 2_1 \alpha 3_1$ collagen arrangement, with prevalence also inconsistent across some taxonomic categories. To date, no chondrichthyan has been shown to possess an α3 (I) chain. Of interest, Kimura [31] found it to be present in the sturgeon *Acipenser transmontanus*, whereas we were unable to find such a sequence for *Acipenser schrenckii* using the online protein search tools.

Additionally, a number of the species known to be tetraploid in our database (e.g. *Oncorhynchus kisutch, O. mykiss, O. tshawytscha, Salmo salar* [59], *Sinocyclocheilus anshuiensis, S. graham* and *S. rhinocerous* [60]) have two sequence variations for most of their type (I) α-chains (labelled in this study as isoforms A and B, differing from one another by approximately 3–10% in any one species), presumably because they have a different copy of the gene that codes for each. The exceptions here are *O. kisutch*, α1; *O. mykiss*, α3; *O. tshawytscha*, α2; *S. graham*, α3; *S. rhinocerous*, α1 (I) and α3 (I), for which we could find just one isoform in each case (electronic supplementary material, table S5),

which may be linked to reported tissue specificity [34], environmentally driven changes in α-chain expression [35], or may simply be absent from the database rather than the species. Of great interest, we have shown that it is possible to visualize both isoforms of each amino acid chain in both a collagen (I) peptide mass fingerprint (electronic supplementary material, figure S1 and table S6) and through liquid chromatography tandem mass spectrometry analyses for modern salmon bone (*Salmo salar*) [28].

# 3. Results and discussion

## 3.1. Collagen (I) phylogenies

### 3.1.1. Topology

Topology was assessed by comparing our concatenated collagen (I) derived trees to the comprehensive genomic phylogenies published by Betancur-R *et al.* [4] and Hughes *et al.* [2], the latter study of which used protein translations to reduce the confounding effect of base-composition heterogeneity, known to be a biasing factor among fish taxa [61]. The topological arrangements yielded by our Bayesian trees under the Dayhoff model display 23% (11 out of 48) of node positions as incongruent with the most recent comprehensive genetically derived tree from Hughes *et al.* [2] (figure 1*a*). In comparison, the two genetically derived trees are highly congruent and, using our limited taxon set, only two nodes are resolved differently between the two genomic trees (figure 1*b*). Node incongruences between collagen (I) and genomic trees occur almost exclusively within the Percomorpha (subdivision Percomorphaceae), which includes over half of all extant teleost diversity and is known for its relatively poor phylogenetic resolution generally [12], due to rapid radiation and diversification in the Cretaceous [2]. The one exception to this is a single-node incongruence within the *Sinocyclocheilus* spp. It is intriguing but not altogether surprising that the vast majority of uncertainties in the collagen-based trees are within the percomorphs, the same taxonomic grouping that has previously shown reduced consensus in genetic-based trees. This suggests that rapid radiation events work to reduce phylogenetic signal in collagen-based phylogenies. The percomorph fishes are divided into nine series (supraordinal groups) that are well supported in recent genomic analyses. Our taxon set includes five of these series: Syngnatharia (e.g. seahorses), Eupercaria (e.g. sticklebacks), Ovalentaria (e.g. cichlids), Carangaria (e.g. jacks, flounders) and Anabantaria (e.g. swamp eels). While these supraordinal groups are monophyletic in the genomic trees (figure 1*b*), they are more dispersed in the collagen (I) trees (figure 1*a*). In particular, the seahorse (Syngnatharia; *Hippocampus comes*) is resolved as sister to all other percomorphs using genomic data but becomes a higher level node, grouped with the Japanese puffer (Euperaria; *Takifugu rubripes*) and the green spotted puffer (Euperaria; *Dichotomyctere nigroviridis*) in the protein trees. The Ovalentaria, however, do appear to retain their monophyletic relationship in the collagen (I) analyses, with the exception of the bicolor damselfish (*Stegastes partitus*; Pomacentridae) that is resolved within a clade comprising other percomorph series (Carangaria, Eupercaria). These observations are consistent with the extensive functional constraints of such an integral and highly abundant protein, where the acceptance of amino acid substitutions may be constrained by the requirement to retain protein structure and function, affecting how the collagen (I) sequence evolves.

Despite the differences between genomic and collagen-based trees, the levels of incongruence between them are relatively low considering the former were produced with protein sequences of just three partial genes, coding for the triple-helical region of collagen (I), as opposed to entire genomes. Hughes *et al.* [2] also found mostly congruent topologies between their genomic and protein trees, but note that the latter were based on 185 096 amino acids as opposed to just 3161 in this study.

### 3.1.2. Tree space visualization

The relative contribution of each α-chain to the combined-partitioned analysis was assessed via a tree space visualization of the posterior estimates using both classical MDS (figure 2*a*) and NMDS (figure 2*b*). Under both MDS methods, we find that partition-specific trees occupy largely non-overlapping regions of tree space. This suggests that each α-chain contains an independent phylogenetic signal. Tree samples from both α1 (I) or α3 (I) partitions occupy larger areas of tree space. This indicates that there is topological uncertainty in the data; trees estimated using only α1 (I) or α3 (I) may be very different from each other. By contrast, α2 (I) trees occupy a much smaller region of tree space; trees estimated using only α2 (I) are more similar to each other. Likewise, trees estimated from the combined-partitioned data occupy a small

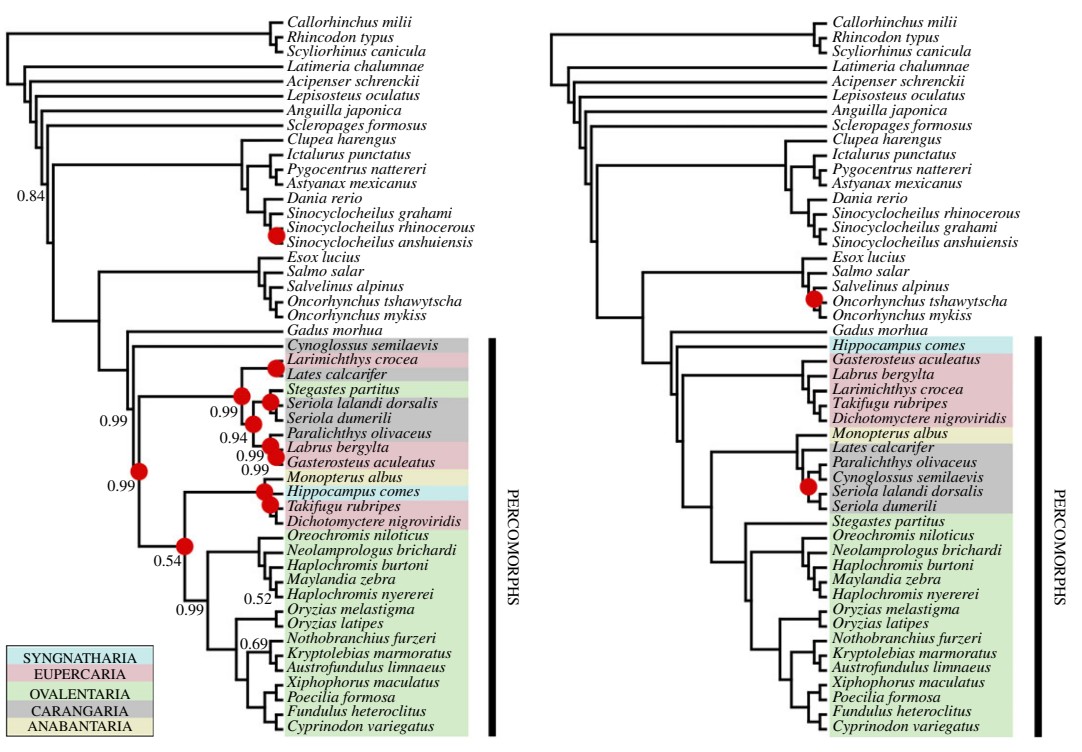

**Figure 1.** Topological comparisons between our concatenated 50% MRC collagen (I) tree (*a*) and published genomic trees ([4], ML; [2], ML) (*b*). Red circles mark incongruent nodes. Node support (posterior probability) is displayed for the collagen (I) tree (*a*). Nodes without support values have a posterior probability of 1. Percomorphs are colour coded to represent supraordinal groups. Our MRC and MCC trees had identical topologies; hence only one is shown here (see electronic supplementary material, Information).

region of tree space. Evidently, combining all three α-partitions minimizes uncertainty inherent in the α1 (I) and α3 (I) partitions. The combined-partitioned trees occupy a unique region of tree space that overlaps only slightly with the a3 (I) region. This suggests that the partitioned data provide hidden support for topologies that are not apparent when analysing each of the α-partitions separately. The combined-partitioned trees and α2 (I) trees are all reasonably close to the Hughes *et al.* [2] tree (mean quartet distance of 19 123.6 and 19 937.65, respectively). Some α3 (I) trees are close to the Hughes *et al.* [2] tree, while others are very distant (mean quartet distance = 23 367.08). Most α1 (I) trees are distant from the Hughes *et al.* [2] tree (mean quartet distance = 30 313.71). Assuming that the comprehensive genomic tree of Hughes *et al.* [2] is close to the true actinopterygian tree, these results suggest that the α2 (I) partition is more congruent with the true tree than either α1 (I) or α3 (I), and that combining the data yields trees that are closer still to the true tree.

### 3.1.3. Molecular clock

Our IGR relaxed clock analysis resolves divergence times that are fairly comparable with genomic-based clock analyses, including the mean node age of the Actinopteri at ~294 million years (Ma), the base of Teleostei at ~212 Ma and the radiation of percomorphs beginning at ~141 Ma (table 1 and figure 3; electronic supplementary material, Information). This suggests that collagen (I) shows fairly clock-like evolution, at least within the Actinopterygii. Due to its highly conserved nature, this is also likely to be the case for vertebrates more generally and is a promising avenue for future research.

### 3.1.4. Collagen (I) sequence evolution

Modelled rates of collagen (I) amino acid sequence evolution for each of the three α-chains were calculated in MrBayes using both a uniform and birth–death clock prior. The results of each clock prior generated the same results, both showing that the α3 (I) amino acid sequence is the quickest evolving with an average of 1 substitution per Ma across the chain of 1062 amino acids (table 2). The second quickest is the α2 (I) chain at a rate of 0.9 substitutions per Ma across 1041 sites, followed by

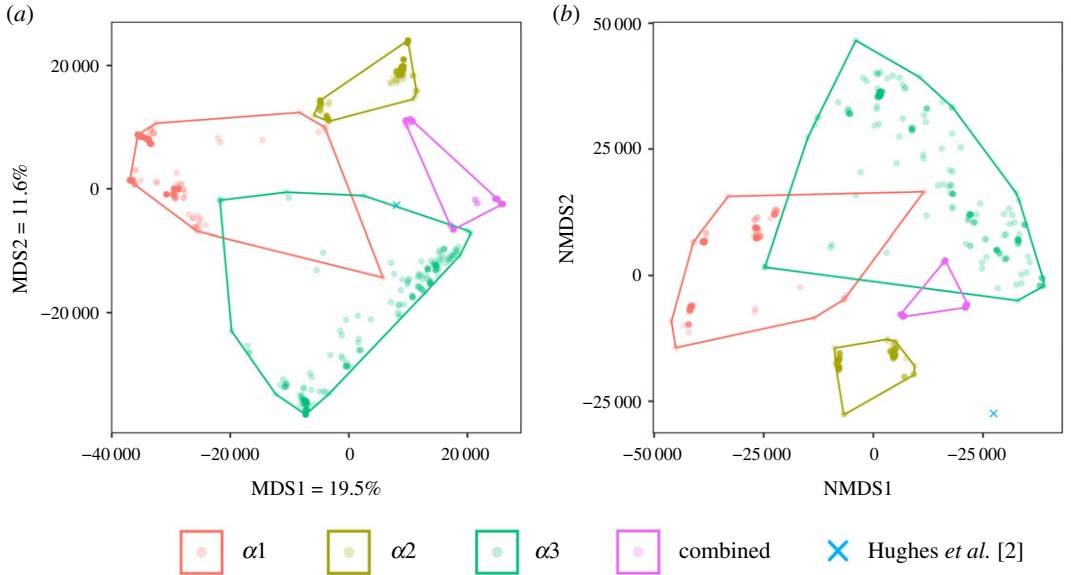

**Figure 2.** Tree space visualization, using classical MDS (*a*) and NMDS (*b*), showing the tree space occupation of partition-specific trees, combined-partitioned trees and the consensus tree of Hughes *et al.* [2]. We sampled 200 post-burnin trees from the posterior sample of each partition-specific topology, as well as 200 combined partition trees. Lighter shades indicate few duplicate trees, darker shades indicate a greater number of duplicate trees. The percentage of variance for each axis is stated on the relevant axis label. Note that there is no percentage of variance associated with individual axes in NMDS.

**Table 1.** Estimated time-calibrated mean ages (to the nearest Ma) of base nodes between Betancur-R *et al.* [4], Hughes *et al.* [2] and this study. We report our age estimates under the uniform and birth–death clock models, respectively. Ages that were not stated are completed with a dash (–); all ages in Hughes *et al.* [2] are estimated from their published phylogeny figures.

| base node | Betancur *et al.* [4] | Hughes *et al.* [2] | this study |
| --- | --- | --- | --- |
| Osteichthyes | 425[b] | — | 428.6/428.9[b] |
| Actinopteri | 350[b] | ~350 | 291.7/293.1 |
| Neopterygii | 323[b] | — | 247.3/248.1[b] |
| Teleostei[a] | 283 | ~250[b] | 213.1/211.1[b] |
| Clupeacephala | 250 | ~220[b] | 201.4/185.6[b] |
| Acanthopterygii | 146 | ~130 | 162.2/158.6 |
| Percomorphaceae | 133 | ~120[b] | 145.5/140.5 |
| Otophysa | 197 | ~128 | 129.0/133.0 |

[a]excluding Lepisosteiformes.
[b]calibrated node.

$\alpha1$ at 0.7 substitutions per Ma across 1058 sites (table 2). These results align broadly with the chain evolution hypothesis as reviewed by Buckley [30], suggesting that a higher observed level of sequence diversity between closely related species, as in $\alpha2$ (I) and $\alpha3$ (I), is likely to indicate a higher rate of sequence evolution. These results also confirm that fish collagen (I) amino acid sequences are the fastest evolving of any vertebrate group, which is of particular importance to disciplines where the presence of evolutionary-driven collagen (I) amino acid substitutions can be interrogated for species identification (e.g. [28,62]). Such relatively rapid sequence evolution may be influenced, in part, by biological reproduction rates. For example, highly prolific mammal species, such as rats, are seen to have a higher rate of amino acid substitution than longer lived mammal species with slower reproductive rates [63]. It is possible that the high fecundity of fishes, in general, may account for such rapid rates of sequence evolution.

Although increasing in availability, collagen (I) sequences are still rare and underrepresented in the field of proteomics. For fishes, the available collagen (I) sequences represent less than 0.2% of the total

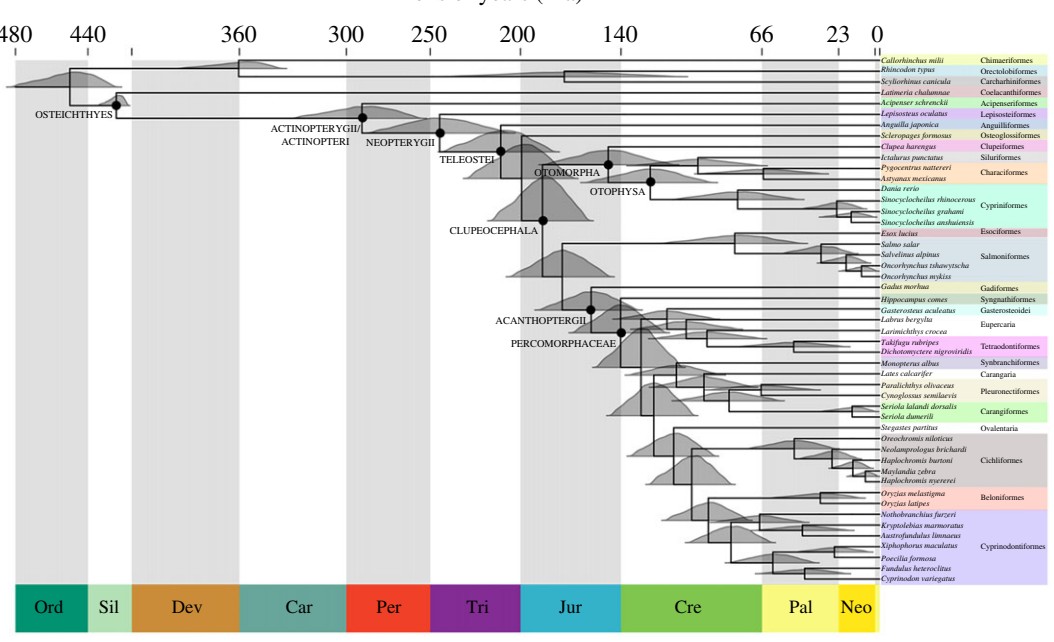

**Figure 3.** Time-calibrated Bayesian phylogeny of Actinopterygii, using a birth–death clock prior, showing 95% HPD confidence intervals as distributions (grey). Classifications were assigned following FishBase [1]. Our time-calibrated Bayesian phylogeny using a uniform clock prior was near-identical and can be found in the electronic supplementary material, Information.

**Table 2.** Modelled rates of collagen (I) amino acid sequence evolution for each of the three α-chains, calculated in MrBayes using a birth–death clock prior. Rates were also modelled using a uniform clock prior, which agreed with the birth–death estimates within 0.00001 substitutions per site per Ma. Outputs show that the $\alpha 3$ (I) sequence evolves at a quicker rate (1 substitution per Ma across the chain), compared to $\alpha 2$ (0.9) and $\alpha 1$ (0.7).

| birth–death clock | | | |
|---|---|---|---|
| partition | rate (=substitutions per site per million years) | number of amino acids in chain | rate per α-chain per million years |
| a1 | 0.00067 | 1058 | 0.7 |
| a2 | 0.00088 | 1041 | 0.9 |
| a3 | 0.00094 | 1062 | 1.0 |
| mean | 0.00083 | 1054 | 0.9 |

number of modern fish species described to science today. While this study represents the forefront of our understanding of collagen-based phylogenetic analysis and sequence evolution rate estimates, the frequency and variety of published collagen (I) sequences is currently its greatest limitation. Even though this primary investigation into collagen (I) amino acid sequence usage for actinopterygian phylogenetics did not contain any extinct fish species, the results of this study suggest that there is value in undertaking such analyses for extinct taxa that have preserved collagen in the bone remains. However, as our data suggest, caution must be taken if analysing fossil percomorphs.

# 4. Conclusion

There are an increasing number of studies using collagen (I) for phylogenetic inference, particularly those that attempt to include extinct vertebrate species using archaeological or palaeontological bone material that can be more challenging for DNA recovery. The comparison of collagen- and genomic-based phylogenetic trees presented here to test the utility of collagen (I) in ray-finned fish systematics shows

that collagen-based phylogenetic trees generate topological arrangements that are 77% congruent with recent genetically derived trees. The discrepancies between the collagen- and gene-based trees appear to occur almost exclusively within the Percomorpha taxonomic subdivision, most likely as a consequence of highly rapid radiation that occurred in the Cretaceous. We do not see incongruences in older tree nodes suggesting that there are more congruent topologies at deeper phylogenetic scales. Of great importance is that our collagen-derived molecular clock trees generate divergence times that are comparable with genomic-based phylogenetic analyses (particularly for more recently diverging nodes) and, despite the highly conserved nature of collagen (I), show the rate of sequence evolution to be an average of 0.9 amino acid substitution for every million years divergence per α-chain—the quickest rate known for any vertebrate group. We conclude that the protein collagen (I) yields phylogenetic signals that are fairly congruent with genomic-level trees both in terms of topology and divergence times. This is promising given they are sourced from just three transcribed genes of approximately 1040 amino acids each, as opposed to entire genomes. Although we do not advocate collagen-based phylogenetics as a replacement for gene-based trees, this study shows collagen (I) as a phylogenetically informative biomolecule that will be of greatest importance when incorporating extinct actinopterygian species into phylogenies via collagen (I) sequencing.

Data accessibility. All sequences, scripts and output files associated with this publication are available from the Dryad Digital Repository: https://doi.org/10.5061/dryad.xgxd254gs. Electronic supplementary material, tables and figures are present in the electronic supplementary material for this study.

The data are provided in the electronic supplementary material [64].

Authors' contributions. V.L.H., J.N.K. and M.B. conceived and designed the experiments; V.L.H. and M.B. retrieved raw data of online sequence databases; V.L.H. arranged and analysed sequences; V.L.H. and M.B. designed the concept behind phylogenetic analyses; J.N.K. designed and wrote the code for Bayesian phylogenies, tree space visualization and molecular clock analysis, with input from V.L.H. V.L.H. and J.N.K. wrote the manuscript, with contributions from M.B.

Competing interests. We declare we have no competing interests.

Funding. Many thanks go to the University of Manchester Dean's Award for scholarship funding to V.L.H. and support from the Royal Society for fellowship funding to M.B. (grant no. UF120473). J.N.K. was funded by ERC grant no. 788203 (INNOVATION) and BBSRC Standard grant no. BB/N015827/1.

Acknowledgements. Many thanks go to Dr Christopher Knight for his guidance and advice during earlier versions of the manuscript, and to Professors Stefano Mariani and Susanne Shultz for reviewing and providing guidance during later versions.

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
