## [Peer Review File · Royal Society Open Science]

Review History

RSOS-201955.R0 (Original submission)

Review form: Reviewer 1

Is the manuscript scientifically sound in its present form?

Yes

Are the interpretations and conclusions justified by the results?

Yes

Is the language acceptable?

Yes

Do you have any ethical concerns with this paper?

No

Have you any concerns about statistical analyses in this paper?

Yes

Recommendation?

Major revision is needed (please make suggestions in comments)

Comments to the Author(s)

Please see attached file (see Appendix A).

Review form: Reviewer 2

Is the manuscript scientifically sound in its present form?

Yes

Are the interpretations and conclusions justified by the results?

Yes

Is the language acceptable?

Yes

Do you have any ethical concerns with this paper?

No

Have you any concerns about statistical analyses in this paper?

No

Recommendation?

Accept with minor revision (please list in comments)

Comments to the Author(s)

In the manuscript "Phylogenetic analyses of ray-finned fishes (Actinopterygii) using collagen type I protein sequences" Harvey et al test the suitability of collagen type I amino acid sequences for phylogenetic estimation of ray-finned fishes. Maximum likelihood and Bayesian phylogenies are reconstructed, using sequencing data available at NCBI and Ensembl, and compared to previously published genomic based topologies. The authors find a high congruence between the genomic phylogenies and the ones obtained using collagen type I protein sequences.

This is an interesting paper that adds new evidence to the use of collagen sequences for species identification and phylogenetic placement, and is worth of publication.

I have a few minor comments, that I list below.

The authors should mention the node support for the different nodes in the collagen trees. These figures are not mentioned anywhere in the text or figures apart from the supplementary information but I think this should be mentioned in the main text.

I think the section "Collagen I sequences: Technical overview" pages 14 and 15 of the manuscript should be part of the methods section as it deals with questions regarding data curation (including collagen chain misidentification) and data parsing and treatment pre-phylogenetic analyses.

Review form: Reviewer 3

Is the manuscript scientifically sound in its present form?

No

Are the interpretations and conclusions justified by the results?

No

Is the language acceptable?

Yes

Do you have any ethical concerns with this paper?

Yes

Have you any concerns about statistical analyses in this paper?

Yes

Recommendation?

Major revision is needed (please make suggestions in comments)

Comments to the Author(s)

The manuscript presents a brief review of collagen (I) diversity among fishes and a phylogenetic analysis to test the utility of collagen (I) molecules to infer phylogeny and estimate divergence times among fish taxa. Since collagen (I) molecules may be retrieved from fossilized remains of long extinct species, collagen-based phylogenies may provide a significant new resource to place fossil species in a phylogenetic framework. A secondary goal of the study is to assess rates of substitution and composition of collagen (I) molecules across the diversity of fishes. The authors use publicly available amino acid sequence data to perform their study.

The test of phylogenetic utility is based on comparison of topologies obtained by the analysis of collagen (I) sequences and topologies obtained previously by analysis of multiple loci or genome-wide data sets. The results show a high level of concordance suggesting that collagen (I) molecules contain useful phylogenetic information. The discrepancies noted were confined to relationships among percomorph taxa, a derived group that has been historically hard to resolve. The time-calibrated tree obtained by analysis of collagen (I) sequences also presented reasonable agreement with dates of divergence previously estimated based using genome-wide data. Overall, the study affirms that collagen (I) molecules may provide useful information to place fossil taxa in a phylogenetic framework. The secondary goal to characterize collagen (I) evolution provided less compelling results and is fraught by methodological shortcomings.

The methods used for phylogenetic inference are generally sound and current, but additional analyses would strengthen the conclusions. In particular, model-based analysis used in the study are based on the Dayhoff model (for Bayesian inference) or on the mtREV24 model (for ML inference), based on previous work or on analysis using MEGA. I suggest additional model-testing using the program IQ-TREE 2 (Minh et al 2020, <https://doi.org/10.1093/molbev/msaa015> that offers a number of advanced models including partitioned models, mixture models, posterior-mean site frequency models, and heterotachy models.

Curating the published sequences for collagen (I) to produce alignments for phylogenetic analysis has been done by eye, especially to distinguish the alpha 1 and alpha 3 orthologs (Technical overview section, starting in line 379). The public databases have mistaken labels for these two variants due to the similarity in name (COL1A1a and COL1A1b, line 397). The authors provide a “rule” to discriminate these variants based on their sequence (K in 9th position of helical domain versus Q in that position). Orthology assessment, a precious concept in molecular evolution and phylogenetics, is best resolved based on phylogenetic analysis. It would have been most revealing to see a gene tree of all collagen (I) molecules to explicitly map the duplication

events in the organismal phylogeny. Some discussion of when these duplications occurred might be offered in light of such gene tree in relation to the whole-genome duplications known to have occurred in vertebrates and in teleosts.

Explicit justification for each calibration point used for time-calibration is missing. Best practice in phylogenetics requires detailed explanation and justification for choosing specific fossil taxa to calibrate nodes in a phylogeny. Minimally, the numbers for the ST function (displayed in Figure S1) need to be explained, and also justified even if Benton and Donoghue provided the original proposal for these numbers. Furthermore, given the relatively modest size of the dataset involved, the use of approximate methods like MCMCTree is not necessary. Full implementation of time-tree analysis provided by MrBayes or BEAST software should be preferred since these methods do not use approximations that are necessary short-cuts when analyzing large genome-wide data. These methods also provide explicit tests for “clock-like” behavior of the analyzed sequences, a much better alternative to just using a crude comparison of estimated divergence dates as an indication for clock-like evolution (Line 292).

Phylogenetic comparative methods (introduced by Felsenstein in 1985) would be necessary to assess compositional differences in collagen (I) sequences among taxa and their distribution among “habitat types” (warm-water or cold-water). The simple t-test employed does not consider phylogenetic structure and is therefore not appropriate to analyze these data. Since this topic seems secondary to the main goal of the paper, I suggest excluding it from a revised version. To adequately address the question “to assess mutation rate in terms of the amino acid sequence composition of species from warm-water (high Pro+Hyp concentration) versus cold-water (low Pro+Hyp concentration) habitats,” more sophisticated model-based methods would be required.

In conclusion, this study is a meaningful contribution to the phylogenetic literature of fishes and provides an interesting resource going forward to incorporate new fossil evidence. One question that was not specifically addressed in the manuscript is whether the collagen (I) data would be particularly useful to place fossil taxa along deeper (unsampled) branches of the fish phylogeny or if it would be best to consider these data more like a barcode (similar to the COI mtDNA sequence) to place fossils close to extant species, or both. It would have been quite illuminating to actually see an example of how collagen (I) data obtained from a fossil fish may be treated and placed into the phylogeny.

Other comments and suggestions follow:

Line 306 How is “mutation rate” estimated? This section is not clear.

Figure 1: it is not clear which topology is being shown in each case (A-D), please specify if this is the topology obtained in the present study or previously published results. Also, support values for clades are not reported (they are shown in the supplementary files) but it would be relevant to see here whether the conflicting nodes received high or low support from analysis of collagen data.

Figure 2: it would be useful to include the Betancur et al and Hughes et al all trees in this analysis to visualize how close (or far) each collagen partition (or concatenated data) tree is from trees estimated on multi-locus or genome-wide data. Drawing ellipses around the clouds of points also may help visualizing these results (see Appendix B).

Figure 3: what classification is used to designate orders shown in the right panel (e.g. “Perciformes”)? At any rate, this figure is somewhat redundant considering that the topologies are shown in Fig 1 and the table 1 has divergence dates, so it is not really necessary and could be omitted.

Figure 4: this figure is somewhat confusing. Are trait values branch lengths or evolutionary rates? Why is the scale showing negative values (blue)? How can a branch length or a substitution rate be negative?

Decision letter (RSOS-201955.R0)

Dear Dr Harvey

The Editors assigned to your paper RSOS-201955 "Phylogenetic analyses of ray-finned fishes (Actinopterygii) using collagen type I protein sequences" have now received comments from reviewers and would like you to revise the paper in accordance with the reviewer comments and any comments from the Editors. Please note this decision does not guarantee eventual acceptance.

Please submit your revised manuscript and required files (see below) no later than 21 days from today's (ie 15-Mar-2021) date. Note: the ScholarOne system will 'lock' if submission of the revision is attempted 21 or more days after the deadline. If you do not think you will be able to meet this deadline please contact the editorial office immediately.

on behalf of Professor Matthew Collins (Associate Editor) and Kevin Padian (Subject Editor)
openscience@royalsociety.org

Associate Editor Comments to Author (Professor Matthew Collins):

Associate Editor: 1

Comments to the Author:

Please see comments in PDF.

Associate Editor: 2

Comments to the Author:

A very interesting exploration of the evolution of a key structural protein in fish, for example linking rates of evolution to water temperature.

Reviewer comments to Author:

Reviewer: 1

Comments to the Author(s)

Please see attached file.

Reviewer: 2

Comments to the Author(s)

In the manuscript "Phylogenetic analyses of ray-finned fishes (Actinopterygii) using collagen type I protein sequences" Harvey et al test the suitability of collagen type I amino acid sequences for phylogenetic estimation of ray-finned fishes. Maximum likelihood and Bayesian phylogenies are reconstructed, using sequencing data available at NCBI and Ensembl, and compared to previously published genomic based topologies. The authors find a high congruence between the genomic phylogenies and the ones obtained using collagen type I protein sequences.

This is an interesting paper that adds new evidence to the use of collagen sequences for species identification and phylogenetic placement, and is worth of publication.

I have a few minor comments, that I list below.

The authors should mention the node support for the different nodes in the collagen trees. These figures are not mentioned anywhere in the text or figures apart from the supplementary information but I think this should be mentioned in the main text.

I think the section "Collagen I sequences: Technical overview" pages 14 and 15 of the manuscript should be part of the methods section as it deals with questions regarding data curation (including collagen chain misidentification) and data parsing and treatment pre-phylogenetic analyses.

Reviewer: 3

Comments to the Author(s)

The manuscript presents a brief review of collagen (I) diversity among fishes and a phylogenetic analysis to test the utility of collagen (I) molecules to infer phylogeny and estimate divergence times among fish taxa. Since collagen (I) molecules may be retrieved from fossilized remains of long extinct species, collagen-based phylogenies may provide a significant new resource to place fossil species in a phylogenetic framework. A secondary goal of the study is to assess rates of substitution and composition of collagen (I) molecules across the diversity of fishes. The authors use publicly available amino acid sequence data to perform their study.

The test of phylogenetic utility is based on comparison of topologies obtained by the analysis of collagen (I) sequences and topologies obtained previously by analysis of multiple loci or genome-wide data sets. The results show a high level of concordance suggesting that collagen (I) molecules contain useful phylogenetic information. The discrepancies noted were confined to

relationships among percomorph taxa, a derived group that has been historically hard to resolve. The time-calibrated tree obtained by analysis of collagen (I) sequences also presented reasonable agreement with dates of divergence previously estimated based using genome-wide data. Overall, the study affirms that collagen (I) molecules may provide useful information to place fossil taxa in a phylogenetic framework. The secondary goal to characterize collagen (I) evolution provided less compelling results and is fraught by methodological shortcomings.

The methods used for phylogenetic inference are generally sound and current, but additional analyses would strengthen the conclusions. In particular, model-based analysis used in the study are based on the Dayhoff model (for Bayesian inference) or on the mtREV24 model (for ML inference), based on previous work or on analysis using MEGA. I suggest additional model-testing using the program IQ-TREE 2 (Minh et al 2020, <https://doi.org/10.1093/molbev/msaa015> that offers a number of advanced models including partitioned models, mixture models, posterior-mean site frequency models, and heterotachy models.

Curating the published sequences for collagen (I) to produce alignments for phylogenetic analysis has been done by eye, especially to distinguish the alpha 1 and alpha 3 orthologs (Technical overview section, starting in line 379). The public databases have mistaken labels for these two variants due to the similarity in name (COL1A1a and COL1A1b, line 397). The authors provide a “rule” to discriminate these variants based on their sequence (K in 9th position of helical domain versus Q in that position). Orthology assessment, a precious concept in molecular evolution and phylogenetics, is best resolved based on phylogenetic analysis. It would have been most revealing to see a gene tree of all collagen (I) molecules to explicitly map the duplication events in the organismal phylogeny. Some discussion of when these duplications occurred might be offered in light of such gene tree in relation to the whole-genome duplications known to have occurred in vertebrates and in teleosts.

Explicit justification for each calibration point used for time-calibration is missing. Best practice in phylogenetics requires detailed explanation and justification for choosing specific fossil taxa to calibrate nodes in a phylogeny. Minimally, the numbers for the ST function (displayed in Figure S1) need to be explained, and also justified even if Benton and Donoghue provided the original proposal for these numbers. Furthermore, given the relatively modest size of the dataset involved, the use of approximate methods like MCMCTree is not necessary. Full implementation of time-tree analysis provided by MrBayes or BEAST software should be preferred since these methods do not use approximations that are necessary short-cuts when analyzing large genome-wide data. These methods also provide explicit tests for “clock-like” behavior of the analyzed sequences, a much better alternative to just using a crude comparison of estimated divergence dates as an indication for clock-like evolution (Line 292).

Phylogenetic comparative methods (introduced by Felsenstein in 1985) would be necessary to assess compositional differences in collagen (I) sequences among taxa and their distribution among “habitat types” (warm-water or cold-water). The simple t-test employed does not consider phylogenetic structure and is therefore not appropriate to analyze these data. Since this topic seems secondary to the main goal of the paper, I suggest excluding it from a revised version. To adequately address the question “to assess mutation rate in terms of the amino acid sequence composition of species from warm-water (high Pro+Hyp concentration) versus cold-water (low Pro+Hyp concentration) habitats,” more sophisticated model-based methods would be required.

In conclusion, this study is a meaningful contribution to the phylogenetic literature of fishes and provides an interesting resource going forward to incorporate new fossil evidence. One question that was not specifically addressed in the manuscript is whether the collagen (I) data would be particularly useful to place fossil taxa along deeper (unsampled) branches of the fish phylogeny

or if it would be best to consider these data more like a barcode (similar to the COI mtDNA sequence) to place fossils close to extant species, or both. It would have been quite illuminating to actually see an example of how collagen (I) data obtained from a fossil fish may be treated and placed into the phylogeny.

Other comments and suggestions follow:

Line 306 How is “mutation rate” estimated? This section is not clear.

Figure 1: it is not clear which topology is being shown in each case (A-D), please specify if this is the topology obtained in the present study or previously published results. Also, support values for clades are not reported (they are shown in the supplementary files) but it would be relevant to see here whether the conflicting nodes received high or low support from analysis of collagen data.

Figure 2: it would be useful to include the Betancur et al and Hughes et al all trees in this analysis to visualize how close (or far) each collagen partition (or concatenated data) tree is from trees estimated on multi-locus or genome-wide data. Drawing ellipses around the clouds of points also may help visualizing these results (see attached figure).

Figure 3: what classification is used to designate orders shown in the right panel (e.g. “Perciformes”)? At any rate, this figure is somewhat redundant considering that the topologies are shown in Fig 1 and the table 1 has divergence dates, so it is not really necessary and could be omitted.

Figure 4: this figure is somewhat confusing. Are trait values branch lengths or evolutionary rates? Why is the scale showing negative values (blue)? How can a branch length or a substitution rate be negative?

===PREPARING YOUR MANUSCRIPT===

If you have been asked to revise the written English in your submission as a condition of publication, you must do so, and you are expected to provide evidence that you have received language editing support. The journal would prefer that you use a professional language editing

service and provide a certificate of editing, but a signed letter from a colleague who is a native speaker of English is acceptable. Note the journal has arranged a number of discounts for authors using professional language editing services (<https://royalsociety.org/journals/authors/benefits/language-editing/>).

===PREPARING YOUR REVISION IN SCHOLARONE===

<https://royalsociety.org/journals/authors/author-guidelines/#supplementary-material> to include a suitable title and informative caption. An example of appropriate titling and captioning

may be found at https://figshare.com/articles/Table_S2_from_Is_there_a_trade-off_between_peak_performance_and_performance_breadth_across_temperatures_for_aerobic_sc_ope_in_teleost_fishes_/3843624.

Author's Response to Decision Letter for (RSOS-201955.R0)

See Appendix C.

RSOS-201955.R1 (Revision)

Review form: Reviewer 3

Is the manuscript scientifically sound in its present form?

Yes

Are the interpretations and conclusions justified by the results?

Yes

Is the language acceptable?

Yes

Do you have any ethical concerns with this paper?

No

Have you any concerns about statistical analyses in this paper?

No

Recommendation?

Accept as is

Comments to the Author(s)

I commend the authors for a thorough revision of their manuscript. I find the current version much improved from the original and the replies from the authors have satisfied my concerns. Congratulations for such a nice contribution!

Decision letter (RSOS-201955.R1)

Dear Dr Harvey,

I am pleased to inform you that your manuscript entitled "Phylogenetic analyses of ray-finned fishes (Actinopterygii) using collagen type I protein sequences" is now accepted for publication in Royal Society Open Science.

on behalf of Prof Kevin Padian (Subject Editor)
openscience@royalsociety.org

Reviewer comments to Author:
Reviewer: 3

Comments to the Author(s)

I commend the authors for a thorough revision of their manuscript. I find the current version much improved from the original and the replies from the authors have satisfied my concerns. Congratulations for such a nice contribution!

Appendix A

This study aims to assess the performance of collagen (I) amino acid sequences for phylogenetic inference and divergence time estimation, and compares the results with the current “gold standards” based on target enrichment / genomic scale datasets. The outline of the approach appears to me to be sound, and as a visual reader, I appreciate the care the authors have taken with the figures.

However, I have some major concerns regarding the details of the methods:

—Most broadly, the structure of the current text makes it unclear whether this is a paper on protein evolution or phylogenetic marker performance. Late in the paper, the authors present overviews of collagen structure and properties (p. 12, lines 305 onward, and p. 14, lines 379 onward), that interrupt the preceding discussion of nodewise age comparisons, and also discuss evolutionary implications based on the topologies they have generated (p. 6, collagen topology). Both of these directions are different from, “is this topology similar to the gold standards, and how close is it.” The back-and-forth makes the text occasionally difficult to read, and I strongly suggest that, if the paper is to be a marker comparison paper, that the authors focus more exhaustively on the metrics specific to this kind of study.

—A major analytical concern I have is with the model of amino acid substitution chosen. The authors explain their justification for using the Dayhoff matrix, but it seems that the Dayhoff matrix appears highly suboptimal for this dataset. I ran an analysis of amino acid model fit using the Java-based program ProtTest (<https://github.com/ddarriba/prottest3>), and generated a ranked list of the model fits under the corrected Akaike Information Criterion (AICc), below. Dayhoff appears over halfway down the list, and mtREV appears much higher, though even at rank 2, it is still outside the AIC weights region that is dominated by mtMam. I would recommend the authors redo analyses under a best-fit model of amino acid substitution. It may be the case that parameter inferences remain largely unchanged, but then it will be for certain inferences by best practices (this goes for the models for the individual chains, as well).

CORRECTED AKAIKE INFORMATION CRITERION

Best model according to AICc: **MtMam+G+F**

Confidence Interval: 100.0

Model	deltaAICc	AICc	AICcw	-lnL
MtMam+G+F	0.00	147924.72	1.00	73803.73
MtREV+G+F	1092.29	149017.01	0.00	74349.88
CpREV+G+F	1999.21	149923.92	0.00	74803.33
JTT+G+F	2324.88	150249.60	0.00	74966.17
RtREV+G+F	2751.81	150676.53	0.00	75179.64

Dayhoff+G	3241.18	151165.89	0.00	75445.15
DCMut+G	3268.96	151193.68	0.00	75459.04
FLU+G+F	3410.76	151335.48	0.00	75509.11
LG+G+F	3484.25	151408.97	0.00	75545.86
WAG+G+F	3680.00	151604.72	0.00	75643.73
JTT+G	3862.64	151787.35	0.00	75755.88
MtArt+G+F	3911.34	151836.06	0.00	75759.40
Dayhoff+G+F	4229.53	152154.25	0.00	75918.50
MtMam+I+F	4248.12	152172.84	0.00	75927.79
DCMut+G+F	4286.45	152211.16	0.00	75946.95
VT+G+F	4625.89	152550.61	0.00	76116.68
MtREV+I+F	5131.79	153056.51	0.00	76369.63
HIVb+G+F	5757.99	153682.71	0.00	76682.73
JTT+I+F	5783.91	153708.63	0.00	76695.69
WAG+G	5972.61	153897.33	0.00	76810.86
Blosum62+G+F	6501.90	154426.62	0.00	77054.68
RtREV+I+F	6562.59	154487.30	0.00	77085.02
CpREV+I+F	6826.16	154750.87	0.00	77216.81
CpREV+G	6837.79	154762.50	0.00	77243.45
Dayhoff+I	7019.85	154944.57	0.00	77334.49
DCMut+I	7039.76	154964.48	0.00	77344.44
LG+G	7126.03	155050.74	0.00	77387.57
JTT+I	7185.91	155110.63	0.00	77417.52
LG+I+F	7222.93	155147.65	0.00	77415.20
VT+G	7354.26	155278.97	0.00	77501.69
WAG+I+F	7653.48	155578.20	0.00	77630.47
Dayhoff+I+F	7911.18	155835.89	0.00	77759.32
DCMut+I+F	7958.20	155882.92	0.00	77782.83
VT+I+F	8088.98	156013.70	0.00	77848.22
RtREV+G	8172.31	156097.03	0.00	77910.72
HIVw+G+F	8340.24	156264.96	0.00	77973.85
FLU+I+F	8749.25	156673.97	0.00	78178.36
MtArt+I+F	8771.84	156696.56	0.00	78189.65
MtMam+F	9437.40	157362.12	0.00	78523.53
Blosum62+I+F	10030.38	157955.10	0.00	78818.92
WAG+I	10065.69	157990.40	0.00	78857.40
FLU+G	10403.83	158328.54	0.00	79026.47
HIVb+I+F	10682.04	158606.76	0.00	79144.75
VT+I	10851.06	158775.78	0.00	79250.09
LG+I	11062.92	158987.64	0.00	79356.02
Blosum62+G	11103.16	159027.88	0.00	79376.14
MtREV+G	11154.50	159079.22	0.00	79401.81
MtREV+F	11334.86	159259.58	0.00	79472.26
CpREV+I	11606.23	159530.95	0.00	79627.68
MtArt+G	11725.15	159649.87	0.00	79687.14
RtREV+I	12220.73	160145.44	0.00	79934.92

MtMam+G	12729.57	160654.29	0.00	80189.35
HIVb+G	12905.55	160830.27	0.00	80277.34
JTT+F	14166.36	162091.08	0.00	80888.01
RtREV+F	14251.71	162176.42	0.00	80930.69
Blosum62+l	14730.79	162655.51	0.00	81189.96
CpREV+F	14949.96	162874.68	0.00	81279.82
LG+F	15192.46	163117.18	0.00	81401.07
Dayhoff+F	15213.62	163138.34	0.00	81411.64
DCMut+F	15275.34	163200.06	0.00	81442.51
MtREV+l	15318.32	163243.04	0.00	81483.72
FLU+l	15387.27	163311.99	0.00	81518.20
WAG+F	15568.45	163493.17	0.00	81589.06
HIVw+l+F	15653.37	163578.09	0.00	81630.42
MtArt+F	15952.53	163877.25	0.00	81781.10
Dayhoff	16025.20	163949.91	0.00	81838.25
DCMut	16057.35	163982.07	0.00	81854.32
VT+F	16477.93	164402.65	0.00	82043.80
JTT	16679.33	164604.05	0.00	82165.32
HIVb+l	17250.32	165175.04	0.00	82449.72
MtArt+l	17336.26	165260.98	0.00	82492.69
MtMam+l	17596.96	165521.68	0.00	82623.04
Blosum62+F	18606.51	166531.23	0.00	83108.09
WAG	18728.23	166652.94	0.00	83189.76
FLU+F	19250.60	167175.31	0.00	83430.13
MtREV	20039.89	167964.61	0.00	83845.59
LG	20059.57	167984.28	0.00	83855.43
RtREV	20200.37	168125.09	0.00	83925.84
VT	20268.62	168193.34	0.00	83959.96
CpREV	20496.84	168421.56	0.00	84074.07
HIVw+G	20533.76	168458.48	0.00	84091.44
MtMam	21018.21	168942.92	0.00	84334.75
HIVb+F	21485.56	169410.28	0.00	84547.62
Blosum62	23919.22	171843.93	0.00	85785.26
MtArt	24630.44	172555.15	0.00	86140.87
HIVw+l	25328.89	173253.60	0.00	86489.00
FLU	25818.67	173743.39	0.00	86734.99
HIVb	29081.66	177006.38	0.00	88366.48
HIVw+F	29575.83	177500.55	0.00	88592.75
HIVw	38570.98	186495.69	0.00	93111.14

Related to the above, it is not clear to me why the authors chose to run two different amino acid models under Bayesian and likelihood analyses, or why the model selection method they used excluded mtMam (and by what information criterion). Both mtREV and mtMam are available options in mrbayes. e.g. aamodelpr=fixed(mtrev).

—p 5, line 143. What, specifically, is “topologies were assessed by direct comparison?” When I see this terminology, I assume “by eye,” as the wording is similar to the old phrasing regarding manual curation of sequence alignments to remove “ambiguous regions.”

—p. 5, lines 173-174. NNI is an outdated search stringency at this point, especially for 60-70 taxa. Please use SPR.

—p. 6, line 184. Why were the trees unrooted, or were outgroups dropped before analyzing the treespace?

—p. 6, line 185. Please elaborate--do you mean Principal Components Analysis, or Principal Coordinates Analysis? This sentence appears to be mixing types. PC in the figures refers to Principal Components, but analysis on a distance matrix suggests PCoA. In the figures, if you are using Principal Coordinates, please abbreviate as PCO1, PCO2 to be consistent with standard abbreviations for the terms.

—p. 6, section. Looking at the Quartet package, it seems less functional than a package like "treespace," which has many functions for plotting and subsetting. See the treespace() function in the package "treespace," and treeDist() for non-PCoA options.

—Figure 2. Please change the labels PC1, PC2 to PCO1, and PCO2. What do these components capture, conceptually? Is PCO1 related to swaps at deeper nodes? Is PCO2 related to instability in percomorphs? Please elaborate.

—p. 15, line 456. This phrasing seems odd. Do you mean, “more congruent topologies at deeper phylogenetic scales?”

—p. 15, lines 459-460. This result seems to me mentioned here for the first time. I do not see it in the abstract, but it might be worth mentioning there.

—p. 15, lines 461-462. If there is a heterogeneous evolutionary rate, might it be worth running a divergence time estimation using BEAST2? I am not familiar with the clock methods used in this manuscript.

—line 39. percomorphs, not percamorphs.

—line 59. If this is a direct quote, please include the pages in the citation.

—line 112. Is “scrutinized” the right word? Perhaps assessed, or analyzed.

—line 293. “Conserved,” or “constrained,” not conservative.

—line 447. “Number,” not quantity. Papers are a discrete count value.

—line 447. “Phylogenetic inference”, or “phylogeny estimation,” not recovery.

Appendix B

Appendix C

Response to Reviewers and Editors

We thank all the reviewers and the editors for their time, care and attention in providing us their comments.

Associate Editor Comments to Author (Professor Matthew Collins):

Associate Editor: 1

Comments to the Author:

- Furthermore, we apply a Bayesian molecular clock approach and compare estimated **divergence** dates with previously published genomic clock analyses.
To:
Furthermore, we apply a Bayesian molecular clock approach and compare estimated divergence dates with previously published genomic clock analyses.

We couldn't find this spelling error in our final version, so we can consider this corrected.

- The custom database was then opened in sequence **aligning** software BioEdit (v.7.2.5)
To
The custom database was then opened in sequence alignment software BioEdit (v.7.2.5)

Corrected.

- It is intriguing but not altogether surprising that the uncertainties in the collagen-based trees are within the same taxonomic region, the Percomorpha, that has previously shown reduced consensus in genetic-based trees, albeit a more enhanced loss of topological consensus in the former (**Figure 1**).
To
It is intriguing but not altogether surprising that the uncertainties in the collagen-based trees are within the same taxonomic region, the Percomorpha, that has previously shown reduced consensus in genetic-based trees, albeit a more enhanced loss of topological consensus in the former (Figure 1).

We couldn't find this spelling error in our final version, so we can consider this corrected.

- Figure 4: Rate tree showing the combined rates analysis for all collagen (I) three amino acid chains ($\alpha 1$ – $\alpha 3$). Lines shaded according to trait values with red representing more rapid sequence evolution compared to blue. Trait values represent the branch lengths (branch-specific substitution rates) as calculated by MCMCtree. Taxa names are shaded according to the **environmental climate**.
*Species without an $\alpha 3$ (I) chain.
To
Figure 4: Rate tree showing the combined rates analysis for all three collagen (I) amino acid chains ($\alpha 1$ – $\alpha 3$). Lines shaded according to trait values with red

representing more rapid sequence evolution compared to blue. Trait values represent the branch lengths (branch-specific substitution rates) as calculated by MCMCtree. Taxa names are shaded according to the **water temperature**.

*Species without an $\alpha 3$ (I) chain.

We have removed Figure 4 from the manuscript.

- In $\alpha 1$ (I) chains, the sequence is present as two distinct tryptic peptides, separated by a lysine **residue** (K) in the ninth position
To
In $\alpha 1$ (I) chains, the sequence is present as two distinct tryptic peptides, separated by a lysine residue (K) in the ninth position;

Corrected.

- As this study represents a highly diverse taxonomic range, **includin** osteichthyans (20 orders)
To
As this study represents a highly diverse taxonomic range, including osteichthyans (20 orders)

We couldn't find this spelling error in our final version, so we can consider this corrected.

- Home page at: www.fishbase.org (12/2019)2019.
To
Home page at: www.fishbase.org (12/2019)2019.

Corrected.

- Danio rerio is often Danio **rario**,

Both versions of 'rario' corrected to 'rerio' – thank you!

Associate Editor: 2

Comments to the Author:

A very interesting exploration of the evolution of a key structural protein in fish, for example linking rates of evolution to water temperature.

Reviewer Comments to Author

Reviewer: 1

Comments to the Author(s)

This study aims to assess the performance of collagen (I) amino acid sequences for phylogenetic inference and divergence time estimation, and compares the results with

the current “gold standards” based on target enrichment / genomic scale datasets. The outline of the approach appears to me to be sound, and as a visual reader, I appreciate the care the authors have taken with the figures.

Many thanks for these supportive comments.

However, I have some major concerns regarding the details of the methods:

—Most broadly, the structure of the current text makes it unclear whether this is a paper on protein evolution or phylogenetic marker performance. Late in the paper, the authors present overviews of collagen structure and properties (p. 12, lines 305 onward, and p. 14, lines 379 onward), that interrupt the preceding discussion of nodewise age comparisons, and also discuss evolutionary implications based on the topologies they have generated (p. 6, collagen topology). Both of these directions are different from, “is this topology similar to the gold standards, and how close is it.” The back-and-forth makes the text occasionally difficult to read, and I strongly suggest that, if the paper is to be a marker comparison paper, that the authors focus more exhaustively on the metrics specific to this kind of study.

We really appreciate this suggestion and have moved the ‘Collagen I sequences: Technical overview’ section to the bottom of the ‘Methods’ section. This was a solution suggested by Reviewer 2 and we feel it vastly improves the flow of the manuscript.

Reviewer 1 also comments here on the section ‘Collagen (I) sequence evolution’. This section has been much reduced and Figure 4 removed. What remains of this section now appears at the end of the manuscript and directly addresses our secondary aim (outlined at the end of the Introduction): “Our secondary objective was to use molecular clock models to assess amino acid substitution rates for each of the collagen (I) $\alpha 1$, $\alpha 2$ and $\alpha 3$ chains in fishes.” We believe we have addressed the ‘back and forth’ issue and we are so pleased that our manuscript is now so much more succinct.

—A major analytical concern I have is with the model of amino acid substitution chosen. The authors explain their justification for using the Dayhoff matrix, but it seems that the Dayhoff matrix appears highly suboptimal for this dataset. I ran an analysis of amino acid model fit using the Java-based program ProtTest (<https://github.com/ddarriba/prottest3>), and generated a ranked list of the model fits under the corrected Akaike Information Criterion (AICc), below. Dayhoff appears over halfway down the list, and mtREV appears much higher, though even at rank 2, it is still outside the AIC weights region that is dominated by mtMam. I would recommend the authors redo analyses under a best-fit model of amino acid substitution. It may be the case that parameter inferences remain largely unchanged, but then it will be for certain inferences by best practices (this goes for the models for the individual chains, as well).

This is a really interesting finding. We have rerun our dataset in PartitionFinder2 using the ‘MrBayes only’ option and testing both linked and unlinked branch lengths. Both branch length options support Dayhoff as the best fitting model for our dataset, above

MTREV, MTMAM and all other models. This is contra to what Reviewer 1 found here which we find puzzling. Presumably different methods of model fitting produce different model best fits. We have edited our manuscript to state that we used PartitionFinder2 to find the most optimal model for our dataset. We have therefore undertaken our analysis under a best-fit model of amino acid substitution, as recommended by PartitionFinder2. The discrepancy between model fitting software is intriguing and warrants further study, this is clearly beyond the remit of our current study. Our Supplementary info contains the output files for PartitionFinder2 for both the linked and unlinked branch lengths.

[See model assessment in PDF]

Related to the above, it is not clear to me why the authors chose to run two different amino acid models under Bayesian and likelihood analyses, or why the model selection method they used excluded mtMam (and by what information criterion). Both mtREV and mtMam are available options in mrbayes. e.g. aamodelpr=fixed(mtrev).

We initially presented both Bayesian and Maximum likelihood phylogenies in order to explore differences in character optimisation. However, the results of these two methods were largely in agreement. To streamline the manuscript, we have decided to remove the ML results. This omission in no way changes the results of our study.

—p 5, line 143. What, specifically, is “topologies were assessed by direct comparison?” When I see this terminology, I assume “by eye,” as the wording is similar to the old phrasing regarding manual curation of sequence alignments to remove “ambiguous regions.”

Thanks for this. We have added text to clarify how the tree topologies were assessed. “Trees were pruned in R to include the maximum number of taxa common to all sets of data (i.e. both collagen and genomic datasets) (n=28). The trees were then drawn in the programme Mesquite (44), and compared in R using the function comparePhylo() in the ape package (43).”

—p. 5, lines 173-174. NNI is an outdated search stringency at this point, especially for 60-70 taxa. Please use SPR.

We have deleted the ML tree as per an earlier comment, so there is now no mention of NNI.

—p. 6, line 184. Why were the trees unrooted, or were outgroups dropped before analyzing the treespace?

The outgroups were removed because all these taxa lack a3 (I) sequences and thus we are unable to estimate a3 (I) partition-specific topologies if these taxa are included (the Bayesian analysis fails to converge).

Rooted vs. unrooted trees can produce different tree to tree distances using some metrics. For example, using the Robinson-Foulds metric, the maximum distance between two unrooted trees is $2(n-3)$, where as for rooted trees it is $2(n-2)$. Given that the root is specified prior to the analysis, rooted tree to tree distances may be considered biased (rooted trees will always share the user specified outgroup). However, in the present study, we computed tree to tree distances using the quartet metric, which measures the number of quartets (unrooted 4 taxon subtrees) that are resolved differently between two trees. The quartet distance between two trees is the same irrespective of whether either is rooted. We have therefore removed this line from the manuscript to avoid ambiguity.

—p. 6, line 185. Please elaborate--do you mean Principal Components Analysis, or Principal Coordinates Analysis? These sentence appears to be mixing types. PC in the figures refers to Principal Components, but analysis on a distance matrix suggests PCoA. In the figures, if you are using Principal Coordinates, please abbreviate as PCO1, PCO2 to be consisted with standard abbreviations for the terms.

Thanks, we have changed these to MDS1 and MDS2, and edited Figure 2 axes.

—p. 6, section. Looking at the Quartet package, it seems less functional than a package like "treespace," which has many functions for plotting and subsetting. See the treespace() function in the package "treespace," and treeDist() for non-PCoA options.

We thank the reviewer to alerting us to the treespace package (Jombart et al 2017). On reviewing this paper, we noted the author's use of Cailliez's transformation, necessary to transform non-euclidean tree to tree matrices into Euclidean distance matrices. We have updated our script to include this important step.

Our preferred tree to tree distance metric is the quartets metric, which has a number of benefits over other distance metrics. In particular, it is less biased by rogue taxa (see Keating et al 2020). The quartet metric is currently not implemented in the aforementioned package. Instead, the custom script we have provided is able to compute quartet distance matrices using the Quartet.Status() function, from the package Quartet (Smith 2019). The reviewer is correct that the Quartet package, by itself, is of limited utility for tree space visualisation. However, the script we have included can be readily modified to implement any tree to tree distance metric, or indeed any multidimensional scaling method. In our modified Figure 2, we have included visualisations using classical multidimensional scaling and nonmetric multidimensional scaling. We have also included our rationale above in the text for this section, and supply our code in the Supplementary information.

—Figure 2. Please change the labels PC1, PC2 to PCO1, and PCO2. What do these components capture, conceptually? Is PCO1 related to swaps at deeper nodes? Is PCO2 related to instability in percomorphs? Please elaborate.

Thanks, we have changed these axes in Figure 2 to MDS1 and MDS2, and edited the text to match as per previous comment.

—p. 15, line 456. This phrasing seems odd. Do you mean, “more congruent topologies at deeper phylogenetic scales?”

Corrected.

—p. 15, lines 459-460. This result seems to me mentioned here for the first time. I do not see it in the abstract, but it might be worth mentioning there.

Corrected – we’ve added this conclusion into the abstract.

—p. 15, lines 461-462. If there is a heterogeneous evolutionary rate, might it be worth running a divergence time estimation using BEAST2? I am not familiar with the clock methods used in this manuscript.

We thank the reviewer for this suggestion. Since our previous submission, we have omitted Figure 4 and amended our divergence time analysis, which has now been run using MrBayes.

—line 39. percomorphs, not percamorphs.

Corrected.

—line 59. If this is a direct quote, please include the pages in the citation.

Added.

—line 112. Is “scrutinized” the right word? Perhaps assessed, or analyzed.

Changed to ‘analysed’.

—line 293. “Conserved,” or “constrained,” not conservative.

Changed to ‘conserved’.

—line 447. “Number,” not quantity. Papers are a discrete count value.

Corrected.

—line 447. “Phylogenetic inference”, or “phylogeny estimation,” not recovery.

Changed to ‘phylogenetic inference’.

Reviewer: 2

Comments to the Author(s)

In the manuscript “Phylogenetic analyses of ray-finned fishes (Actinopterygii) using collagen type I protein sequences” Harvey et al test the suitability of collagen type I amino acid sequences for phylogenetic estimation of ray-finned fishes. Maximum likelihood and Bayesian phylogenies are reconstructed, using sequencing data available at NCBI and Ensembl, and compared to previously published genomic based topologies. The authors find a high congruence between the genomic phylogenies and the ones obtained using collagen type I protein sequences.

This is an interesting paper that adds new evidence to the use of collagen sequences for species identification and phylogenetic placement, and is worth of publication. I have a few minor comments, that I list below.

Many thanks for these supportive comments.

The authors should mention the node support for the different nodes in the collagen trees. These figures are not mentioned anywhere in the text or figures apart from the supplementary information but I think this should be mentioned in the main text.

We have added all node support values to our two Bayesian trees (MRC, MCC) in the new version of Figure 1, and we think this greatly improves our figure and the manuscript. We have edited the caption for Figure 1 to reference the node support values. Figure 1 is described in detail in the Results and Discussion section under ‘Topology’, and we now supply all MrBayes output files in the Supplementary, which includes the node support values across all our trees.

I think the section “Collagen I sequences: Technical overview” pages 14 and 15 of the manuscript should be part of the methods section as it deals with questions regarding data curation (including collagen chain misidentification) and data parsing and treatment pre-phylogenetic analyses.

We really appreciate this suggestion and have moved the ‘Collagen I sequences: Technical overview’ section to the bottom of the ‘Methods’ section.

Reviewer: 3

Comments to the Author(s)

The manuscript presents a brief review of collagen (I) diversity among fishes and a phylogenetic analysis to test the utility of collagen (I) molecules to infer phylogeny and estimate divergence times among fish taxa. Since collagen (I) molecules may be retrieved from fossilized remains of long extinct species, collagen-based phylogenies may provide a significant new resource to place fossil species in a phylogenetic framework. A secondary goal of the study is to assess rates of substitution and composition of collagen (I) molecules across the diversity of fishes. The authors use publicly available amino acid sequence data to perform their study.

The test of phylogenetic utility is based on comparison of topologies obtained by the analysis of collagen (I) sequences and topologies obtained previously by analysis of multiple loci or genome-wide data sets. The results show a high level of concordance suggesting that collagen (I) molecules contain useful phylogenetic information. The discrepancies noted were confined to relationships among percomorph taxa, a derived group that has been historically hard to resolve. The time-calibrated tree obtained by analysis of collagen (I) sequences also presented reasonable agreement with dates of divergence previously estimated based using genome-wide data. Overall, the study affirms that collagen (I) molecules may provide useful information to place fossil taxa in a phylogenetic framework.

Excellent summary.

The secondary goal to characterize collagen (I) evolution provided less compelling results and is fraught by methodological shortcomings.

The methods used for phylogenetic inference are generally sound and current, but additional analyses would strengthen the conclusions. In particular, model-based analysis used in the study are based on the Dayhoff model (for Bayesian inference) or on the mtREV24 model (for ML inference), based on previous work or on analysis using MEGA. I suggest additional model-testing using the program IQ-TREE 2 (Minh et al 2020, <https://doi.org/10.1093/molbev/msaa015> that offers a number of advanced models including partitioned models, mixture models, posterior-mean site frequency models, and heterotachy models.

We have revised our model fitting using Partition Finder2. These new results support Dayhoff over mtREV24 under both linked or unlinked parameters. The output files are available in the Supplementary information.

The reviewer's suggestions for investigating heterotachy, mixture models etc. are intriguing but we believe they are beyond the remit of our current study. We followed the reviewer's suggestion to rerun the clock analyses in MrBayes and we have therefore opted to focus on MrBayes for all analyses for consistency. Unfortunately, the IQ-Tree programme is also completely new to us, and another reason for prioritising Bayesian estimation is because we have much greater expertise and prior publications in it.

Curating the published sequences for collagen (I) to produce alignments for phylogenetic analysis has been done by eye, especially to distinguish the alpha 1 and alpha 3 orthologs (Technical overview section, starting in line 379). The public databases have mistaken labels for these two variants due to the similarity in name (COL1A1a and COL1A1b, line 397). The authors provide a "rule" to discriminate these variants based on their sequence (K in 9th position of helical domain versus Q in that position). Orthology assessment, a precious concept in molecular evolution and phylogenetics, is best resolved based on phylogenetic analysis. It would have been most revealing to see a

gene tree of all collagen (I) molecules to explicitly map the duplication events in the organismal phylogeny. Some discussion of when these duplications occurred might be offered in light of such gene tree in relation to the whole-genome duplications known to have occurred in vertebrates and in teleosts.

Thank you for this comment, it is insightful, but we feel the creation of a gene tree, as the reviewer suggests, this is outside of the scope of this study as it justifies a study in its own right. Our manuscript did not specifically aim to research the origin of the $\alpha 3$ duplication event, for that we refer to the works of Kimura, 1992 and Morvan-Dubois et al., 2003. If the presence/absence of the $\alpha 3$ chain mapped clearly onto a phylogenetic tree (either genomic or proteomic) then we would have been able to make a statement about the location of the duplication event in the genetic lineage of actinopterygians. However, with relatively little collagen (I) protein sequences available, we feel there is not currently enough evidence to make this statement. Nor unfortunately do we feel we have the expertise to create gene trees to map these duplication events. We have written in some depth in the 'Collagen (I) sequences: Technical overview' section, $\alpha 3$ chains are limited but not universal to teleosts, with some teleosts missing this chain (e.g. *Clupea harengus* and some cyprinids) – but these species do not form a monophyletic clade and thus there is no clear image yet of when in the lineage the $\alpha 3$ chain evolved. This observation alone brings new knowledge of the prevalence of the $\alpha 3$ chain, given that very little other work has been done in this area since the two articles above. Separately, the whole genome duplication events in Salmonidae and some Cyprinidae are our inferred reasons why these species have two versions of each collagen (I) α -chain that differ from one another by about 30%. We can see multiple evidence of both versions of each chain in collagen (I) peptide mass fingerprints in salmon (Supplementary Figure S2). After reviewing our text on these subjects, we are not entirely clear about what else can be added succinctly, further to what we have already included. We are open to more suggestions for this point, particularly if we are missing a trend or valid discussion point.

Explicit justification for each calibration point used for time-calibration is missing. Best practice in phylogenetics requires detailed explanation and justification for choosing specific fossil taxa to calibrate nodes in a phylogeny. Minimally, the numbers for the ST function (displayed in Figure S1) need to be explained, and also justified even if Benton and Donoghue provided the original proposal for these numbers.

We have added a table to the Supplementary with all the calibration details. The same information is also included on the MrBayes nexus files, which are available in the Supplementary information.

Furthermore, given the relatively modest size of the dataset involved, the use of approximate methods like MCMCTree is not necessary. Full implementation of time-tree analysis provided by MrBayes or BEAST software should be preferred since these methods do not use approximations that are necessary short-cuts when analyzing large genome-wide data. These methods also provide explicit tests for “clock-like” behavior of

the analyzed sequences, a much better alternative to just using a crude comparison of estimated divergence dates as an indication for clock-like evolution (Line 292).

We have run the clock analyses in MrBayes as per the suggestion of Reviewer 3 using both a uniform and a birth-death clock prior. Both methods produced identical trees and near identical age estimates and clock rates. All raw data is available in the Supplementary information.

Phylogenetic comparative methods (introduced by Felsenstein in 1985) would be necessary to assess compositional differences in collagen (I) sequences among taxa and their distribution among “habitat types” (warm-water or cold-water). The simple t-test employed does not consider phylogenetic structure and is therefore not appropriate to analyze these data. Since this topic seems secondary to the main goal of the paper, I suggest excluding it from a revised version. To adequately address the question “to assess mutation rate in terms of the amino acid sequence composition of species from warm-water (high Pro+Hyp concentration) versus cold-water (low Pro+Hyp concentration) habitats,” more sophisticated model-based methods would be required.

We appreciate this input here and agree. We have removed the sections of the manuscript that aim to assess rates of collagen (I) evolution in fishes in relation to Pro+Hyp concentration. This includes the deletion of the t-test, amino acid composition calculations and Figure 4, as well as editing the aims in the Introduction section and the reference to Figure 4 in the Discussion. We agree that more sophisticated models are required here to truly assess this relationship and is justified for a separate study.

In conclusion, this study is a meaningful contribution to the phylogenetic literature of fishes and provides an interesting resource going forward to incorporate new fossil evidence. One question that was not specifically addressed in the manuscript is whether the collagen (I) data would be particularly useful to place fossil taxa along deeper (unsampled) branches of the fish phylogeny or if it would be best to consider these data more like a barcode (similar to the COI mtDNA sequence) to place fossils close to extant species, or both. It would have been quite illuminating to actually see an example of how collagen (I) data obtained from a fossil fish may be treated and placed into the phylogeny.

We thank the reviewer for this comment and useful insight. For this scoping study (assessing the utility of collagen (I) phylogenetic signal in general) we decided not to include any fossil taxa because of a number of variables this would introduce, namely sequence coverage. Collagen (I) sequences from extinct taxa are built from bottom-up proteomic analysis incorporating LC-MS/MS and probability matching for each peptide in the sequence. This means that in best case scenarios, we can hope to achieve ~70% sequence coverage for a given taxa (depending on how related it is to other taxa in the database). This is one of the limitations in proteomics, which is well versed in the literature. Whilst it will be extremely valuable to trial this in future studies, we opted to only include full collagen (I) sequences for our taxa so as to truly assess the worth of

collagen-based phylogenies without introducing variables such as sequence coverage. Furthermore, this project unfortunately did not have the resources available to undertake LC-MS/MS analysis. We do hope that future studies will investigate the addition of fossil species, and we hope our study can be the gateway by providing the methods for this. We have mentioned in several locations about the worth of this future work, including in the Abstract, Discussion and Conclusions.

Other comments and suggestions follow:

Line 306 How is “mutation rate” estimated? This section is not clear.

In our revised analyses, we have calculated mutation rate as follows: The mean clockrate (in substitutions per site per million years) was obtained from the .pstat file output by mrbayes, as were the relative partition rates. Multiplying the mean clockrate by the relative partition rate gives the partition rate in substitutions per site per million years. Multiplying this number by the number of sites in the partition gives the mutation rate (in substitutions per million years). We have amended the manuscript so that our mutation rate calculation is explicit.

Figure 1: it is not clear which topology is being shown in each case (A-D), please specify if this is the topology obtained in the present study or previously published results. Also, support values for clades are not reported (they are shown in the supplementary files) but it would be relevant to see here whether the conflicting nodes received high or low support from analysis of collagen data.

Thank you. We have included node values in Figure 1 for our protein trees and included the following in the caption: “Node support values are displayed for collagen (I) trees (A and B).”

Figure 2: it would be useful to include the Betancur et al and Hughes et al all trees in this analysis to visualize how close (or far) each collagen partition (or concatenated data) tree is from trees estimated on multi-locus or genome-wide data.

Great suggestion, we have amended our treespace visualisation to include the Hughes et al. tree.

Drawing ellipses around the clouds of points also may help visualizing these results (see attached figure).

We have amended our tree space visualisation plot so that convex hulls envelop each cloud of trees. This improves the visualisation and we thank the reviewer for this suggestion.

Figure 3: what classification is used to designate orders shown in the right panel (e.g. “Perciformes”)? At any rate, this figure is somewhat redundant considering that the

topologies are shown in Fig 1 and the table 1 has divergence dates, so it is not really necessary and could be omitted.

Thanks for pointing this out. We have edited a few of the classifications so that they now align with those present on Fishbase, thus 'Perciformes' has now been changed to the relevant series within (i.e. Ovalentaria). This figure depicts the clock-like evolution of collagen (I) protein sequence and thus we feel it is a nice visual addition for the manuscript. We have updated the caption to include reference to FishBase for the classifications.

Figure 4: this figure is somewhat confusing. Are trait values branch lengths or evolutionary rates? Why is the scale showing negative values (blue)? How can a branch length or a substitution rate be negative?

We have deleted Figure 4.